**Data Availability Statement:** All relevant data are within the paper.

**Funding:** The author(s) received no specific funding for this work.

# Pre hospital delay and its associated factors in acute myocardial infarction in a developing country

**Ishmum Zia Chowdhury**[1]*, **Md. Nurul Amin**[2], **Mashhud Zia Chowdhury**[3], **Sharar Muhib Rahman**[1], **Mohsin Ahmed**[4], **F. Aaysha Cader**[3]

**1** BIRDEM General Hospital, Dhaka, Bangladesh, **2** Ibrahim Cardiac Hospital and Research Institute, Dhaka, Bangladesh, **3** Department of Cardiology, Ibrahim Cardiac Hospital and Research Institute, Dhaka, Bangladesh, **4** Department of Cardiology, National Institute of Cardiovascular Diseases, Dhaka, Bangladesh

* ishmumchow@gmail.com

## Abstract

### Background

Early revascularization and treatment is key to improving clinical outcomes and reducing mortality in acute myocardial infarction (AMI). In low- and middle-income countries such as Bangladesh, timely management of AMI is challenging, with pre-hospital delays playing a significant role. This study was designed to investigate pre-hospital delay and its associated factors among patients presenting with AMI in the capital city of Dhaka.

### Methods

This retrospective cohort study was conducted on 333 patients presenting with AMI over a 3-month period at two of the largest primary reperfusion-capable tertiary cardiac care centres in Dhaka. Of the total patients, 239(71.8%) were admitted in the National Institute of Cardiovascular Diseases, Dhaka and 94(28.2%) at Ibrahim Cardiac Hospital & Research Institute, Dhaka Data were collected from patients by semi-structured interview and hospital medical records. Pre-hospital delay (median and inter-quartile range) was calculated. Statistical significance was determined by Chi-square test. Multivariate logistic regression analysis was done to determine the independent predictors of pre-hospital delay.

### Results

The mean age of the respondents was 53.8±11.2 years. Two-thirds (67.6%) of the respondents were males. Median total pre-hospital delay was 11.5 (IQR-18.3) hours with median decision time from symptom onset to seeking medical care being 3.0 (IQR: 11.0) hours. Nearly half (48.9%) of patients presented to the hospital more than 12 hours after symptom onset. On multivariate logistic regression analysis, AMI patients with absence of typical chest pain [OR 5.21; (95% CI: 2.5–9.9)], diabetes [OR: 1.7 (95% CI: 1.0–2.9)], residing/staying > 30 km away from nearest hospital at the time of onset [OR: 4.3(95% CI = 2.3–7.2)] and belonged to lower and middle class [OR: 1.9(95% CI = 1.0–3.5)] were significantly associated with pre-hospital delays.

**Competing interests:** The authors have declared that no competing interests exist.

## Conclusion

Acute myocardial infarction (AMI) patients with atypical chest pain, diabetes, staying far away from nearest hospital and belonged to lower and middle socioeconomic strata were significantly associated with pre-hospital delays. The findings could have immense implications for improvements about timely reaching of AMI patients to the hospital within the context of their sociodemographic status and geographic barriers of the city.

## Background

Coronary artery disease (CAD) constitutes 15.9% of all deaths, making it the most common cause of death worldwide [1]. CAD is particularly prevalent in South Asia, with estimates from the Global Burden of Disease Study suggesting that the South Asian region will have more individuals with atherothrombotic cardiovascular disease than any other region in the days ahead [2]. The management of acute myocardial infarction (AMI) is extremely time-sensitive, with a delay in treatment of AMI being associated with increased mortality and morbidity [3]. The beneficial effect of fibrinolytic therapy was evident among patients at highest risk, including the elderly with proportional mortality reduction being significantly greater in patients treated within 2 h compared to those treated later (44% [95% CI 32, 53] vs 20% [15, 25]; p = 0.001) [4]. Early pharmacological or interventional reperfusion decreases mortality of ST-segment elevation myocardial infarction (STEMI). De Luca and associates showed 1 year mortality risk increased by 7.5% for each 30-minute delay in primary percutaneous coronary intervention (PCI) [3]. Furthermore, in non- ST-segment elevation myocardial infarction (NSTEMI) patients with a Global Registry of Acute Coronary Events (GRACE) score of more than 140, undergoing a coronary angiogram within 12 hours after admission was associated with lower risk of ischemic outcomes at 180 days [5].

Updated global cardiovascular society guidelines all recommend that reperfusion therapy is indicated in all patients with symptoms of ischemia of within a 12-hour duration and persistent ST-segment elevation for STEMI [6, 7]. An early invasive approach is also advocated in NSTEMI [8]. The 12 hour window for STEMI is particularly relevant in low and middle income countries (LMIC), where treatment delays [9, 10] due to lack of coordination between facilities, access to interventional cardiology facilities and catheterization laboratories contributed to poorer outcomes [11–14]. Several factors were found associated with increased time from symptom onset to treatment delays in AMI patients. Living in rural areas, hard road travel (poor road conditions or high traffic volume on the road) and lack of transport availability were reasons for prolonged pre-hospital treatment delays in several studies [15–18]. Mujtaba and colleagues [19] in a recent study in Pakistan observed that > 20% of the AMI patients attended the hospital after 6 hours of symptom onset with median pre-hospital time being 120 minutes (interquartile range: 229). The delay was more common among patients aged 41 to 65 years and among females. Factors that cause delayed presentation in hospital were misinterpretation, misdiagnosis, lack of transportation and financial constraint; of these, misdiagnosis act as a significant determinant of delay (p < 0.05). Financial constraint of the patients and the lack of a national insurance plan in many LMICs needs patients to pay out of pocket for cardiac reperfusion therapies which resulted delay in care or no access to care at all [20, 21].

There have been few studies exploring the pre-hospital delay of AMI presentations and its associated factors in Bangladesh. One study has recently been conducted in Northern city of Rajshahi [22] and another in Southern city of Chittagong [23]. On Online search, no study was

found conducted in Dhaka, the capital city of Bangladesh. While several factors for delays were identified in these studies, it remains uncertain if they necessarily reflect those associated with patients presenting to specialized tertiary care hospitals in the capital city of Dhaka. The present study investigating the pre-hospital delay in patients with AMI was carried out in two specialized tertiary cardiac care centers in Dhaka to address this research gap in the previous studies.

## Methodology

### Study type and location

This retrospective cohort study was conducted at two of the largest primary reperfusion-capable cardiac hospitals in Dhaka, namely 1) National Institute of Cardiovascular Diseases (NICVD), Dhaka and 2) Ibrahim Cardiac Hospital and Research Institute (ICHRI), Dhaka. NICVD is the premier cardiac care centre of the government sector, serves patients at a significantly lower cost than centres in the private sector, and is visited by all classes of people. ICHRI is a tertiary cardiac care centre within the private sector, patronized by the more affluent sections of the society, as the cost of all services are borne by the patients themselves. Therefore, in choosing these two different hospitals, we ensured that we received patients from all strata of the society. The study hospitals were termed as index hospital in our study.

### Sample size

The sample size for this study was determined by using the formula: $n = \frac{z^2 p(1-p)}{d^2}$ where z = z value for 95% confidence interval, p = estimated prevalence of pre-hospital delay of more than 12 hours and d = precision time of error for the estimated prevalence. The proportion of patients with pre-hospital delay was derived from a previous study done in Chittagong, the Southern part of Bangladesh city was 62.7%. Thus, z = 1.96, p = 0.627 and d = 0.0627 (10% of p), calculated sample size was 228. Therefore, we included 333 patients presenting with AMI of which 239 (71.8%) were admitted in NICVD and 94 (28.2%) at ICHRI.

### Inclusion and exclusion criteria

All consecutive patients who presented to ICHRI/ NICVD with AMI within the period of November 2019 to January 2020 and who consented to participate, were included. AMI was diagnosed according to criteria stipulated in the 4[th] universal definition of MI [24] and included patients with STEMI and NSTEMI. Patients who could not recall the events and those who received thrombolysis at a different hospital before presenting to ICHRI/NICVD were excluded.

### Ethical clearance

The study protocol was approved by the Ethical Review Committees of both ICHRI and NICVD. Verbal consent was taken from each participant in presence of the ward doctors on duty and the participant had the right to withdraw from the study at any time during the study period. Confidentiality of participants was strictly maintained.

### Dependent variable and independent variables

*The total pre-hospital delay* was the main dependent (outcome) variable. *Total pre-hospital delay* was defined as the time between the onset of symptoms of MI and time of arrival at the designated hospitals' emergency room (ER), considering guideline recommended time cut-

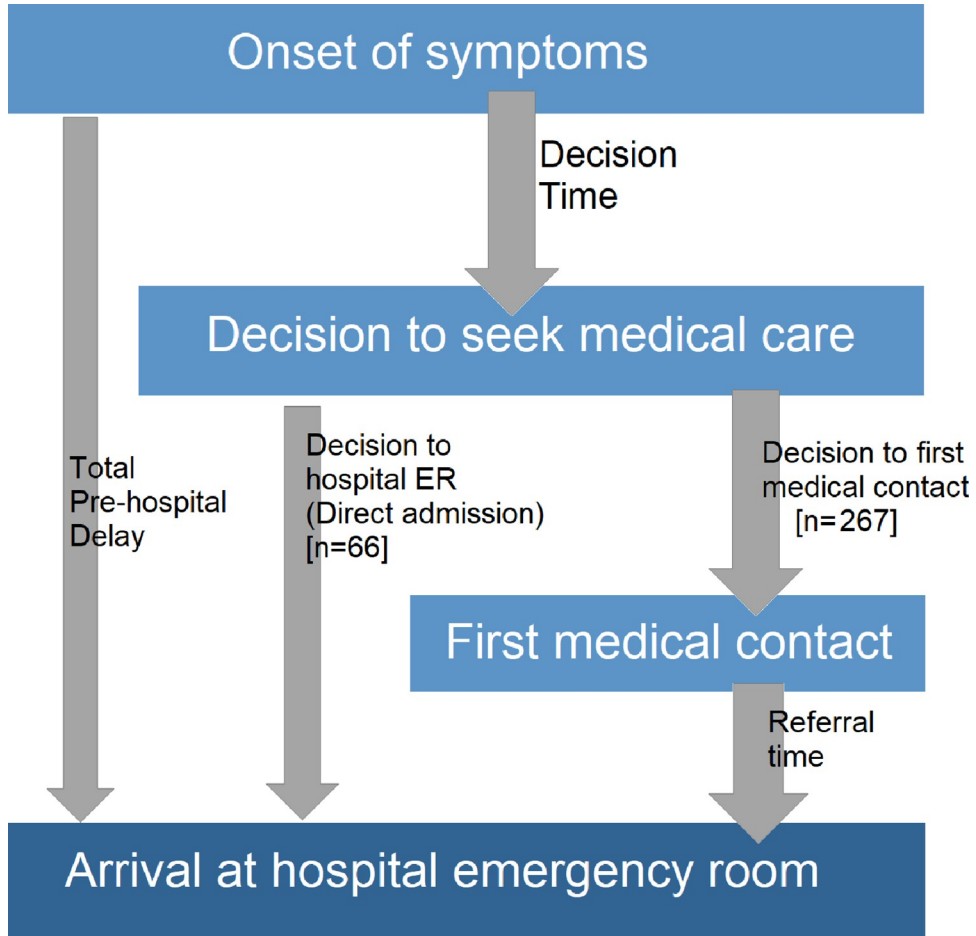

**Fig 1. Working definitions for time intervals related to pre-hospital delay.**

offs in recommendations for reperfusion therapy of STEMI [6, 7]. The pre-hospital delay was further divided into *"Decision time"* and *"Time from decision to hospital ER"*. *"Decision time"* was the time between the onset of patient's symptoms and the time at which a decision was made to seek medical care. *"Time from decision to hospital ER"* is the time starting from a decision to go to hospital up to arrival at the hospital ER. This included time spent at any previous medical contacts and commute to the hospital. (Fig 1). Independent variables included age, sex, literacy, socioeconomic status, presence of typical chest pain, risk factors for CAD, prior history of CAD, first medical contact (FMC), distance from home and type of MI.

## Working definitions

**Socioeconomic status** was categorized by monthly income: patients were asked about their occupation and approximate monthly income during the interview. A monthly income of more than 30,000 BDT (USD 357) were denoted as higher income, between 15,000 BDT (USD 178.5) and 30,000 BDT (USD 357) as middle income and below 15,000 BDT (USD 178.5) as low income.

## Date collection, processing and statistical analysis

Data were collected by means of a pre-tested structured interview, supplemented by medical records. The patients were interviewed when they were clinically stable following treatment/

revascularization and data were recorded on the questionnaire. Sociodemographic characteristics and history from symptom onset were obtained during interview, whereas clinical data were obtained from the patients' hospital charts and electronic medical records. Data analysis was done by Statistical Package for the Social Sciences (SPSS) Version 25.0 (IBM). While continuous variables were expressed as median values with inter quartile range (IQR), categorical data were expressed as frequencies with percentages. Some quantitative variables like age and distance of the index hospital from home were dichotomized for univariate analysis. Univariate analysis was done using Chi-Square ($\chi^2$) test with Odds ratio (OR) to estimate the risk of having the outcome for a particular factor or characteristic. Variables found to be significantly associated with the outcome (pre-hospital) delay in univariate analysis were first subjected to Hosmer and Lemeshow Model-fit Test for Multivariate logistic (Binary logistic) regression analysis. After adjustment for confounding variables by Binary logistic regression analysis, the variables remained to be significantly associated with the outcome variable, were considered as independent predictors.

## Results

### Demographic characteristics, key risk factors, and clinical presentation

A total of 333 patients were studied, of whom two-thirds (67.6%) were males with male to female ratio being 2:1. Demographic and basic clinical characteristics are shown in Table 1. The mean age was 53.8±11.2 years (range 22–90 years). Over 60% presented with STEMI. Over three quarters of the total number of patients (76.9%) patients presented with typical chest pain as compared with 23.1% who presented with atypical symptoms. About 60% had diabetes, 60.1% had hypertension, 40.5% had a history of current/ previous smoking and 55% had family history of CAD. Nearly one-third (31.5%) of the patients were located within 10 km from hospital, 29.1% were at 10–50 km distance, while 39.3% were beyond 50 km distance at the onset of their symptoms (Range: 0.8–417 km). Around three-quarters (72.1%) received some form of preliminary anti-ischemic treatment from the referring clinic before arriving at a tertiary centre. Among them, 56.2% received dual anti-platelet therapy (DAPT) loading dose, 48.9% received sublingual nitroglycerine and 22.8% received low molecular weight heparin.

Overall, only 19.8% of patients presented directly to a tertiary cardiac centre as their FMC (first medical contact) (Table 1). Median total pre-hospital delay was 11.5 (IQR:18.3) hours with median decision time from symptom onset to seeking medical care being 3.0 (IQR:11.0) hours. About 70% of patients had presented to other hospital or clinic first, before eventually arriving at a tertiary centre for definitive care. Among the patients who have had previous medical contact before presenting to ICHRI/NICVD, 144 (53.9%) patients were referred by a general practitioner or community clinic from outside Dhaka city and 123 patients (46.1%) were referred from a different hospital within city areas.

Overall, only 19.8% of patients presented directly to a tertiary cardiac centre as their FMC (First Medical Contact) (Table 1). Median total pre-hospital delay was 11.5 (IQR: 18.3) hours with median decision time from symptom onset to seeking medical care being 3.0 (IQR: 11.0) hours. About 70% of the patients had presented to a single other hospital or clinic from onset of symptoms, before eventually arriving at a tertiary centre for definitive care. Among the patients who had previous medical contact before presenting to ICHRI/ NICVD, 144 (53.9%) patients were referred by a general practitioner or community clinic from outside Dhaka city and 123 patients (46.1%) were referred from a different hospital within city limits.

**Table 1. Demographic and basic clinical characteristics.**

| Characteristics | Number | Frequency |
|---|---|---|
| **Age** | | |
| <35 years | 12 | 3.6 |
| 35–50 years | 101 | 30.3 |
| 51–70 years | 194 | 58.3 |
| Above 70 years | 26 | 7.8 |
| **Sex** | | |
| Male | 225 | 67.6 |
| Female | 108 | 32.4 |
| **Employment status** | | |
| Employed | 159 | 47.7 |
| Unemployed | 174 | 52.3 |
| **Literacy** | | |
| Illiterate | 44 | 13.2 |
| Up to Class 5 | 110 | 33.0 |
| Up to Class 10 | 64 | 19.2 |
| Up to Class 12 | 44 | 13.2 |
| Undergraduate | 55 | 16.5 |
| Post-graduate | 16 | 4.8 |
| **Socioeconomic Status** | | |
| Lower income | 113 | 33.9 |
| Middle income | 132 | 39.6 |
| Higher income | 88 | 26.4 |
| **MI symptom type** | | |
| Typical chest pain | 256 | 76.9 |
| No chest pain | 77 | 23.1 |
| **Self-perception and interpretation of symptoms by patient** | | |
| Gastroenteric/Peptic Ulcer Disease | 179 | 53.8 |
| Muscular | 33 | 9.9 |
| Respiratory | 26 | 7.8 |
| Angina | 77 | 23.1 |
| None | 18 | 5.4 |
| **Risk factors for CAD** | | |
| Diabetes | 199 | 59.8 |
| Hypertension | 200 | 60.1 |
| Smoker/Tobacco | 135 | 40.5 |
| Family History of CAD | 183 | 55.0 |
| **Prior history of CAD** | | |
| None | 214 | 64.3 |
| Chronic Coronary Syndrome | 105 | 31.5 |
| Prior PCI | 12 | 3.6 |
| Prior CABG | 2 | 0.6 |
| **Presentation chronology at tertiary hospital ER** | | |
| First contact | 66 | 19.8 |
| Second contact | 231 | 69.4 |
| Third contact | 36 | 10.8 |
| **Mode of transportation** | | |
| Ambulance/EMS | 146 | 43.8 |

(*Continued*)

**Table 1.** (Continued)

| Characteristics | Number | Frequency |
|---|---|---|
| Private transport | 85 | 25.5 |
| Public transport | 102 | 30.6 |
| **Treatment received prior to arrival at tertiary hospital** | | |
| None | 93 | 27.9 |
| Dual Antiplatelet Therapy loading | 187 | 56.2 |
| Sublingual Nitroglycerine | 163 | 48.9 |
| Low Molecular Weight Heparin | 76 | 22.8 |

## Pre-hospital delays

The distribution of the total pre hospital delay was skewed with a median of 11.5 (IQR-18.3) hours (Table 2). Of the total respondents, 163 patients (48.9%) presented to the hospital more than 12 hours after symptoms onset and the rest 170 (51.1%) within 12 hours of the symptoms. Median decision time from symptom onset to seeking medical care was 3.0 (IQR-11.0) hours. Median time from decision to arrival at hospital ER was 5.0 (IQR-8.0) hours (Figs 2–4). The time from decision to arrival at hospital ER was shorter for those that directly presented to the tertiary hospitals as compared with patients who were referred through a general practitioner (GP), community clinic or a district hospital [2.0 (IQR-2.0) vs. 6.0 (IQR-9.5) hours respectively.

## Factors associated with pre hospital delay

Table 3 shows the factors associated with outcome (pre hospital delay) between those presenting within and beyond 12 hours of symptom onset, with odds ratios (OR) in univariate analysis. Pre-hospital delay was significantly longer in females than males [OR:1.9; (95% CI: 1.2–3.1)], although age did not influence on pre-hospital delay. Respondents from lower and middle-income households were at 2.6 (95% CI: 1.5–4.1) times higher risk of having pre-hospital delay. Patients with level of education below 10th grade tend to be associated with pre-hospital delay with odds of having the condition being 2.3 (95% CI: 1.4–3.5) times higher. Diabetics were more prone to have pre-hospital delay than their non-diabetic counterparts with risk of having the condition 2.1 (95% CI: 1.3–3.3) times greater in the former group than that in the latter group. Patients presenting with atypical chest pain had significantly greater delays than those with typical chest pain with odds of having delay being much higher in the former cohort [5.8 (95% CI = 3.1–10.5)]. Significantly longer pre-hospital delays were also noted in patients located beyond 30 km from the hospital at symptom onset [3.9 (95% CI = 2.5–6.1)] (Table 3).

Analyses of association between diabetes and chest symptoms demonstrated that majority of the patients with atypical chest pain (84.4%) had diabetes as compared 52.3% of the patients

**Table 2. Total pre-hospital delay, decision time and time from decision to reach hospital ER.**

| Statistics of pre-hospital delay | Decision Time (hours) | Time from home to NICVD/ICHRI (hours) | Total time to receive treatment (hours) |
|---|---|---|---|
| Mean (SD) | 10.7(15.7) | 9.4(14.9) | 20.3(22.4) |
| Median (IQR) | 3(11.0) | 5(8.0) | 11.5(18.3) |
| Skewness | 2.2 | 3.3 | 1.9 |

*IQR = Interquartile Range.

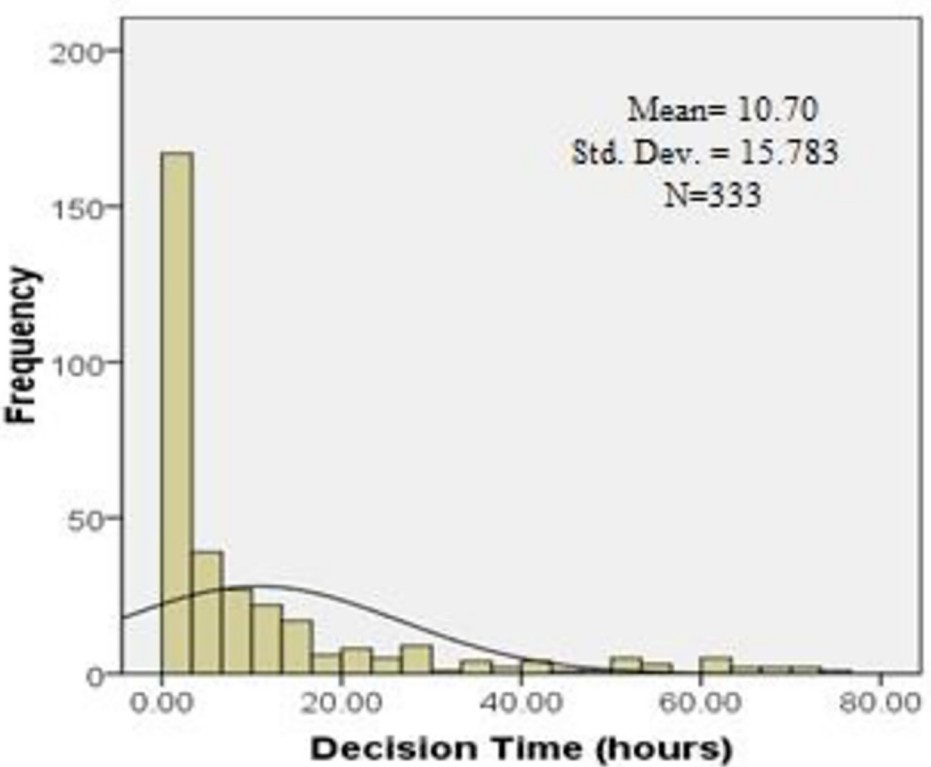

**Fig 2. Histogram showing distribution of decision time with right skewness.**

with typical chest symptoms with risk of having atypical chest pain in diabetics being 4.9 (95% CI: 2.5–9.5) times higher than those without diabetics (Table 4).

Table 5 demonstrates the binary logistic regression analysis of odds ratios for factors associated with pre-hospital delay. The variables found statistically significant in univariate analysis (sex, socioeconomic status, level of education, diabetes mellitus, presence of typical chest pain and distance from the hospital) were all directly entered into the model-fit test first for binary logistic regression analysis. Hosmer and Lemeshow goodness-of-fit test demonstrated that the model was a good-fit-model which could correctly predict the outcome of interest (pre-hospital delay) in 69.3% of the patients (p = 0.081) with overall correct prediction capability of the model being 80.0%. Regression analysis revealed that patients with absence of typical chest pain, residing/staying > 30 km from the index hospital at time of symptom onset, having diabetes and belonged to lower- and middle- socio-economic class were independently associated with pre-hospital delay with odds of having the condition in these cohorts being 4.9 (95% CI: 2.5–9.9), 4.3 (95% CI: 2.3–7.2), 1.7 (95% CI: 1.0–2.9) and 1.9 (95% CI: 1.0–3.5) respectively.

## Discussion

The main aim of this study was to identify predictors of pre-hospital delay for AMI patients in Dhaka, the capital of Bangladesh. We conducted the study at two hospitals: NICVD, a state-run hospital, provides services to a huge volume of patients at a negligible cost and ICHRI is a private hospital requiring patients to finance the whole course of their treatment. Combined, these two hospitals represent a patient population covering all socioeconomic strata of the city. Being two of the largest cardiac tertiary hospitals in the country, this study includes patients

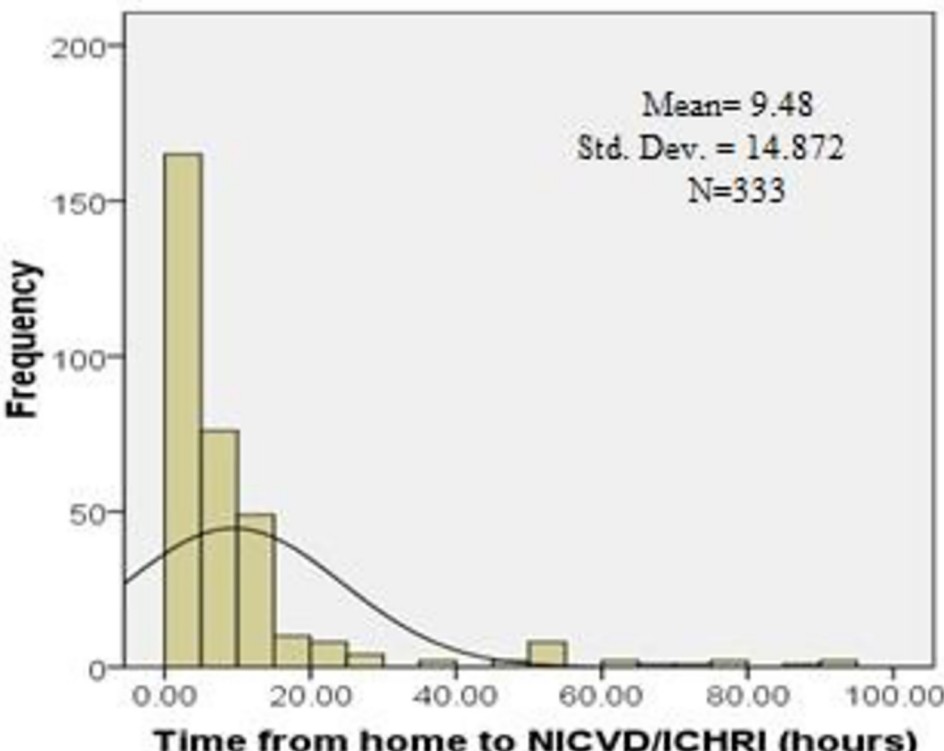

**Fig 3. Histogram showing distribution of time to NICVD/ICHRI with right skewness.**

coming from various cities of Bangladesh, reflecting the contemporary status of the city's cardiac care. There is scarce data on trends and factors responsible for pre-hospital delays in seeking cardiovascular care in Dhaka. This study is unique from prior studies investigating pre-hospital delays in the country for many reasons: 1) the capital city is home to a large population (21,741,000) [25] of various socioeconomic statuses, some of whom can only afford cardiac care in a state hospital, while more affluent people opt for self-financed care in private hospitals. Besides, the capital city receives patients not only from within the city boundaries but across the country, for some of whom definitive cardiac care is only first administered when they reach a tertiary centre.

The median pre hospital delay in our study population was 11.5 (IQR-18.3) hours and the median decision time was 3.0 (IQR-11.0) hours. While the "decision time" is almost entirely attributed to patient factors, the time taken from decision to arrival at NICVD/ICHRI depends on various factors such as referral time, mode of transport, traffic, etc. A previous study by Rafi, et al [22] in Northern Bangladesh revealed a median pre hospital delay of 9 (IQR-13) hours with a median decision time of 2.0 (IQR-1.0) hours. Compared with our study, factors contributing to these slightly lower pre-hospital delays in their study could be attributed to the fact that it was conducted in Northern Bangladesh, where traffic is not as problematic as in the capital city of Dhaka. Furthermore, the hospital at which the study was conducted was a tertiary care centre directly serving the catchment area, which is smaller than that of the present study, and with fewer referrals from across the country, in comparison to those in our study.

Analysis of the factors contributing to pre-hospital delay demonstrated that patients with atypical chest symptoms, residing/staying > 30 km from the index hospital at symptom onset, being belonged to lower and middle income socio-economic class and diabetes were

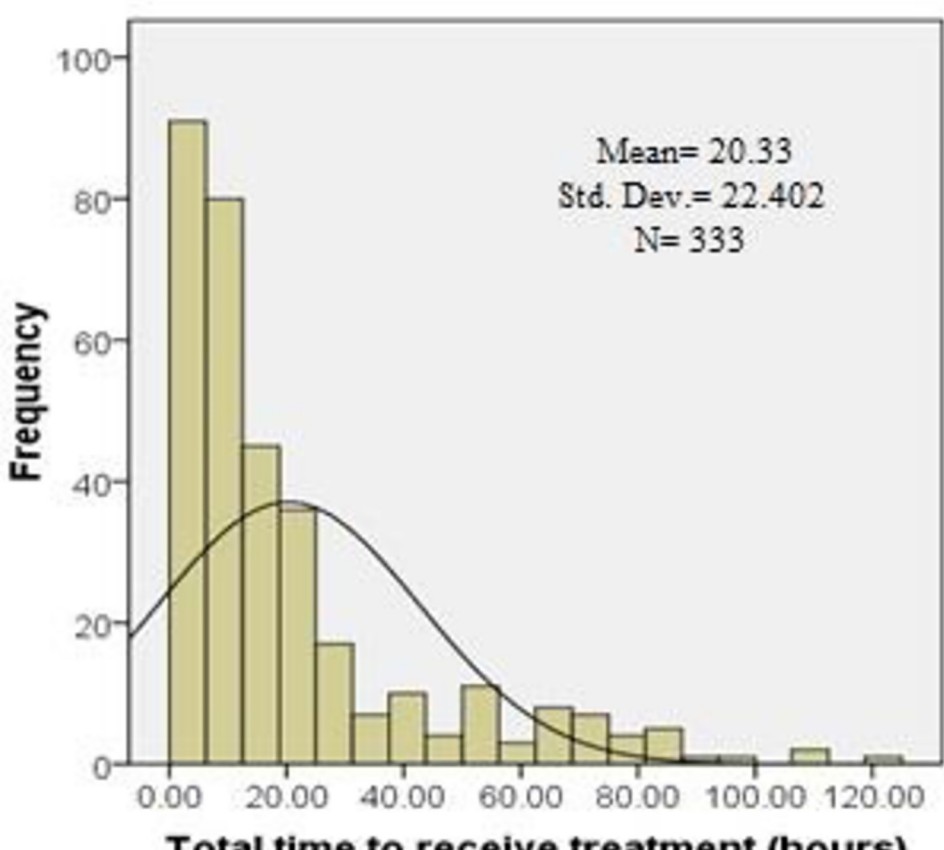

**Fig 4. Histogram showing distribution of time to receive treatment with right skewness.**

independent predictors of pre-hospital delay with odds of having the condition in these cohorts being 4.9 (95% CI: 2.5–9.8), 4.5 (95% CI: 2.3–7.5), 1.9 (95% CI: 1.0–3.5) and 1.7 (95% CI: 1.0–2.9) respectively. Similar to our study, Rafi, et al [22] also found diabetes, lower socio-economic conditions, absence of typical chest pain and greater distance between site of symptoms onset to hospital to be significantly associated with pre-hospital delay. Similar observations were reported in a study by Das et al [23]. They found a mean delay of 6.8 ± 3 hours for those presenting early (as < 12 hours), and a mean delay of 37.8 ± 25.1 hours for late presenters. In addition to distance and diabetes, they found greater delays in those with elderly and those who misinterpreted symptoms for peptic ulcer disease (PUD). Our study results consistent with those from neighbouring LMIC, such as India [15, 26], who also have high pre-hospital delays, and contrasts with high income countries, such as Sweden [27], Australia [28], United States of America [29] and Poland [30] showing a significantly lower pre-hospital delay in patients with AMI.

There are paradoxical finding in literature regarding sex differentials of pre hospital delay. A study conducted in Iran by Farshidi et al [31] and another from Chittagong, Bangladesh [23] showed no significant differences in pre hospital delay between male and female AMI patients. Alternately, in a multinational registry reported by Bugiardini et al [32] and another study conducted in Sweden by Lawesson et al [33] showed that women had significantly longer pre-hospital delays in AMI presentation. In our study, while univariate analysis showed sex as

**Table 3. Factors associated with pre-hospital delay.**

| Demographic, clinical characteristics & geographic barriers | Delay | | Odds ratio | *P-value |
|---|---|---|---|---|
| | > 12 hrs (n = 163) | ≤ 12 hrs (n = 170) | 95 CI of OR | |
| **Sex** | | | | |
| Female | 65(39.9) | 43(25.3) | 1.9(1.2–3.1) | 0.004 |
| Male | 98(60.1) | 127(74.7) | | |
| **Age years** | | | | |
| >50 | 112(68.7) | 108(63.5) | 1.3(0.8–1.9) | 0.318 |
| ≤50 | 51(31.3) | 62(36.5) | | |
| **Socioeconomic Status** | | | | |
| Lower & middle class | 134(82.2) | 111(65.3) | 2.6(1.5–4.1) | <0.001 |
| Upper class | 29(17.8) | 59(34.7) | | |
| **Level of education** | | | | |
| Below 10th grade | 92(56.4) | 62(36.5) | 2.3(1.4–3.5) | <0.001 |
| 10th grade and above | 71(43.6) | 108(63.5) | | |
| **Diabetes** | | | | |
| Present | 112(68.7) | 87(51.2) | 2.1(1.3–3.3) | 0.001 |
| Absent | 51(31.3) | 83(48.8) | | |
| **Hypertension** | | | | |
| Present | 91(55.8) | 109(64.1) | 0.7(0.5–1.1) | 0.123 |
| Absent | 72(44.2) | 61(35.9) | | |
| **Smoker/Tobacco use** | | | | |
| Yes | 71(43.6) | 64(37.6) | 1.3(0.8–1.9) | 0.272 |
| No | 92(56.4) | 106(62.4) | | |
| **Family History IHD** | | | | |
| Present | 97(59.5) | 86(50.6) | 1.4(0.9–2.2) | 0.102 |
| Absent | 66(40.5) | 84(49.4) | | |
| **MI symptom type** | | | | |
| No Chest Pain Atypical | 61(37.4) | 16 (9.4) | 5.8(3.1–10.5) | <0.001 |
| Chest Pain Typical | 102(62.6) | 154 (90.6) | | |
| **Previous experience of IHD, chronic stable angina, PCI or CABG** | | | | |
| Present | 61(37.4) | 58(34.1) | 1.6(0.7–1.8) | 0.529 |
| Absent | 102(62.6) | 112(65.9) | | |
| **Presentation at NICVD/ICHRI** | | | | |
| Direct presentation to ICHRI/NICVD | 129(80.1) | 136(80.0) | 1.0(0.6–1.7) | 0.977 |
| Referred from other centers | 32(19.9) | 34(20.0) | | |
| **Transportation used** | | | | |
| Private/Public Transport | 96(58.9) | 91(53.5) | 1.2(0.8–1.9) | 0.324 |
| EMS/Ambulance | 67(41.1) | 79(46.5) | | |
| **Distance from NICVD/ICHRI** | | | | |
| > 30 km | 108(66.3) | 57(33.5) | 3.9(2.5–6.1) | <0.001 |
| ≤ 30 km | 55(33.7) | 113(66.5) | | |

*Chi-squared $\chi^2$ Test was done analyze the data; figures in the parentheses denote corresponding.

a significant factor, it emerged non-significant on multivariate analysis, indicating sex as a potential confounder for the pre-hospital delay.

As Bangladesh is a developing country, only a negligible percentage of the population has adequate health insurance coverage. In major private institutions offering standard cardiac care, all treatments received by a patient are entirely self-financed, including access to

**Table 4. Association of diabetes with chest symptoms.**

| Diabetes | Chest Pain | | Odds ratio 95 CI of OR | *P-value |
|---|---|---|---|---|
| | Atypical n = 77 | Typical n = 256 | | |
| Present | 65 (84.4%) | 134 (52.3%) | 4.9(2.5–9.5) | < 0.001 |
| Absent | 12 (15.6%) | 122 (47.7%) | | |

emergency medical service (EMS), which possibly factors in the delays in decision time in patients with AMI, which is particularly reflected in the significant differences in pre-hospital delay times across different socioeconomic statuses. Patients with a lower income of less than 30,000 BDT (~357 USD) per month may delay the decision to seek help due to financial reasons for which they have a significantly high pre hospital delay, which is consistent with another study [34]. This is further compounded by possibly a lack of health-seeking behavior owing to a lack of awareness of the time-sensitive nature of cardiac care, and also a dependency on other family members to make key decisions, which largely stems from a general cultural tendency to involve multiple parties in decision-making. Concurrently, patients who have studied beyond the 10th grade showed a significantly lower decision time, possibly resulting from increased knowledge and awareness. Lesser education level has also shown to increase the delay in an Australian study as well [28]. Our study showed that those with diabetes mellitus had the significantly longer decision times. Diabetes Mellitus is not only a major risk factor of coronary artery disease, it also results in silent MI [35] and more atypical symptoms, all of which could contribute to delays in recognizing symptoms, and hence delays in presentation. A large percentage (53.8%) of the patients in our study attributed their symptoms to PUD or acid reflux. Patients who presented with chest pain presented to the hospital relatively earlier than patients who didn't have chest pain as their symptom. This is concurrent with most studies [22, 27].

In contrast to studies in high income countries [36], the mode of transportation to hospital had no significant effect on the pre hospital delay in our study. In high income countries, it has been shown that EMS-transported patients had significantly shorter delays in symptom onset to arrival time in hospitals. However, within Dhaka city limits and indeed in neighboring

**Table 5. Factors associated with pre-hospital delay in multivariate logistic regression.**

| Factors | Univariate Analysis | | Multivariate Analysis | |
|---|---|---|---|---|
| | OR (95% CI of OR) | p-value | OR (95% CI of OR) | p-value |
| **Sex** | 1.9(1.2–3.1) | 0.004 | 1.7(0.9–1.7) | 0.099 |
| FemaleR vs. Male | | | | |
| **Socioeconomic Status** | 2.6(1.5–4.1) | <0.001 | 1.9(1.0–3.5) | 0.036 |
| Lower & Middle incomeR vs. Higher income | | | | |
| **Literacy Status** | 2.3(1.4–3.5) | <0.001 | 1.6(0.9–2.9) | 0.084 |
| Below 10th gradeR vs. Above 10th grade | | | | |
| **DM** | 2.1(1.3–3.3) | 0.001 | 1.7(1.0–2.9) | 0.043 |
| PresentR vs Absent | | | | |
| **MI symptom type** | 5.8(3.1–10.5) | <0.001 | 4.9(2.5–9.9) | <0.001 |
| No chest painR vs Chest pain | | | | |
| **Distance from hospital** | 3.9(2.5–6.1) | <0.001 | 4.3(2.3–7.2) | <0.001 |
| >30 kmR vs ≤30 km | | | | |

R = Reference case.

suburbs, persistent traffic and the lack of emergency lanes preclude any benefit that may be derived from ambulance transportation; furthermore, an ambulance dispatch might even result in further delays considering a lack of organized EMS infrastructure and the time factored in to reach the patient. Additionally, few ambulances are equipped with trained staff that can administer DAPT, LMWH or thrombolytics with appropriate monitoring; as such many patients opt for personal vehicles and more easily available taxi services to reach hospitals. The time delay from FMC also depends on the distance from patient location to FMC: two-thirds (66.3%) of the patients who had delayed presentation reached the index hospitals from a location beyond 30 km as compared to (33.5%) of the patients who reached the hospital within 12 hours.

Although we did not assess clinical outcomes, a delayed presentation of MI is associated with an increased in-hospital mortality according to several studies [37, 38]. Reperfusion therapy is the key strategy to reduce mortality in AMI but its benefit is time dependent, and a 12-hour cut-off is particularly important in case of STEMI [6, 39]. Thus, reduction in the pre hospital delay will reduce mortality in patients with AMI.

This is a study that reflects the contemporary and realistic trends of pre-hospital delay in seeking cardiac care in a LMIC setting. It also highlights the multiple underlying issues of a healthcare system that need to be addressed in order to improve parameters and ensure timely cardiac care. The absence of a competent and well-trained EMS system remains a huge problem compromising timely cardiac care. This is further compounded by the absence of a functional referral system that is structured in a manner that at the very least ensures preliminary care and reperfusion for patients, prior to being transferred for tertiary care.

This study also highlights the variations in FMC sought and tendency of the general public to not adhere to referral systems. Whether this is a general lack of trust in the referral hierarchy, possibly stemming from an absence of fully-equipped treatment for AMI in many non-tertiary centres, or a matter of convenience in directly accessing a specialist, is a matter of debate. Alternatively, the habits of visiting local GPs instead of presenting to hospital ER, reflects a need for public health messaging and education of the general public on appropriate self-presentations for acute medical care. This can be done by locally and socially appropriate, community-targeted campaigns and interventions to increase the knowledge and awareness about the disease and its symptoms, and encourage a healthcare-seeking behavior among the general population. In the currently existing set-up, there is an additional need of adequate training of GPs and non-specialist doctors to promptly administer essential medicines in the management of AMI. Progressive modifications of healthcare infrastructure, with particular focus in improving emergency medical care, training and developing a more competent cohort of paramedics and ambulances, and implementing a proper and well-adhered to referral system, are important factors that need to be considered in decreasing pre-hospital delays in cardiac care.

## Strengths and limitations

Patients' symptoms, diagnoses and other clinical information were obtained from electronic medical records that reduced the risk of errors. Although this study was conducted in two of the largest cardiac hospitals in the country with a large turn over and covered a large portion of urban and peri-urban population, the findings do not necessarily represent those of entire reference population of the country. As > 12-hour delay is a sensitive issue for AMI patients, we dichotomized the delay in ≤ 12 and > 12 hours to see what factors could contribute to > 12 hours delay. However, dichotomizing the hours of delay and doing logistic regression have lost some information on understanding quantitative hours of delay. It could be better

modeled with an over dispersed Poisson regression or negative binomial regression analysis. Besides, the study also did not involve the patients who could not make it to the hospital or died within a few hours of arrival. So, caution should be exercised to generalize the findings to reference population.

## Conclusion

This present study investigating pre-hospital delays in the presentations of AMI patients in the Bangladeshi capital city of Dhaka revealed that the presence of diabetes, lower socioeconomic status, absence of chest pain and a distance from hospital greater than 30 km were significantly associated with increased pre-hospital delays in presentation. Educating people, particularly of low socioeconomic class having diabetes and residing far away from the cardiac care hospitals about AMI symptoms and importance of early presentation to hospital with concurrent improvements of the existing referral system and EMS will go a long way to reduce delays.

## Author Contributions

**Conceptualization:** Ishmum Zia Chowdhury, Mashhud Zia Chowdhury, F. Aaysha Cader.

**Data curation:** Ishmum Zia Chowdhury.

**Formal analysis:** Ishmum Zia Chowdhury, Md. Nurul Amin, F. Aaysha Cader.

**Investigation:** Ishmum Zia Chowdhury, Sharar Muhib Rahman.

**Methodology:** Ishmum Zia Chowdhury, Md. Nurul Amin, Sharar Muhib Rahman, F. Aaysha Cader.

**Project administration:** Ishmum Zia Chowdhury, Mashhud Zia Chowdhury, Mohsin Ahmed, F. Aaysha Cader.

**Resources:** Mashhud Zia Chowdhury.

**Software:** Ishmum Zia Chowdhury, Md. Nurul Amin, Sharar Muhib Rahman, F. Aaysha Cader.

**Supervision:** Md. Nurul Amin, Mashhud Zia Chowdhury, Mohsin Ahmed, F. Aaysha Cader.

**Validation:** Md. Nurul Amin, Mashhud Zia Chowdhury, Mohsin Ahmed, F. Aaysha Cader.

**Writing – original draft:** Ishmum Zia Chowdhury, Sharar Muhib Rahman, F. Aaysha Cader.

**Writing – review & editing:** Ishmum Zia Chowdhury, Md. Nurul Amin, Mashhud Zia Chowdhury, Mohsin Ahmed, F. Aaysha Cader.

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
