## [Decision Letter · Decision Letter 0]

12 May 2021

PONE-D-21-11112

Pre-hospital delay and its associated factors in Acute Myocardial Infarction in a developing country

PLOS ONE

Dear Dr. Chowdhury,

Thank you for submitting your manuscript to PLOS ONE. After careful consideration, we feel that it has merit but does not fully meet PLOS ONE’s publication criteria as it currently stands. Therefore, we invite you to submit a revised version of the manuscript that addresses the points raised during the review process.

Although all reviewers and editors found that this manuscript has merit and addresses a significant clinical problem, the reviewers raise some important issues. These comments include data presentation and statistical analysis (reviewer #1 and #2), discussion to be improved (reviewer #1 and #2), insufficient data on patients’ background, confounders, and quantitative hours of delay (reviewer #2), and sex differences (reviewer#2). In addition, among 333 patients, the number of patients who visited NICVD or ICHRI should be provided separately.

We look forward to receiving your revised manuscript.

Kind regards,

Michinari Nakamura, MD

Academic Editor

PLOS ONE

Journal Requirements:

2. Please provide additional details regarding participant consent.

In the ethics statement in the Methods and online submission information, please ensure that you have specified what type you obtained (for instance, written or verbal, and if verbal, how it was documented and witnessed).

If your study included minors, state whether you obtained consent from parents or guardians.

If the need for consent was waived by the ethics committee, please include this information.

Reviewers' comments:

Reviewer's Responses to Questions

**Comments to the Author**

1. Is the manuscript technically sound, and do the data support the conclusions?

Reviewer #1: Yes

Reviewer #2: Partly

2. Has the statistical analysis been performed appropriately and rigorously? 

Reviewer #1: Yes

Reviewer #2: No

3. Have the authors made all data underlying the findings in their manuscript fully available?

Reviewer #1: Yes

Reviewer #2: Yes

4. Is the manuscript presented in an intelligible fashion and written in standard English?

Reviewer #1: Yes

Reviewer #2: Yes

5. Review Comments to the Author

Reviewer #1: OVERALL SUMMARY

This study presents conclusions for a cross sectional study conduct in a low-income country examining pre-hospital delays for patients seen for cardiovascular disease outcomes adjusting for potential confounders. The work is insightful and thought provoking. See below for comments from my review.

INTRODUCTION

The authors do not discussion findings from previous studies examining pre-hospital delay for cardiovascular outcomes.

Paragraphs 2 and 3 can be combined as they are talking about the same thought. After shortening the combined paragraph more can be added which presents results from actual scientific studies as opposed to just medial guidelines.

The introduction is not sufficient as it stands. The authors need to use past studies to make a case for the issue they are studying and point out the gaps in previous research that their research addresses.

METHODS

The authors need to define mid to lower socio-economic class populations. What income level is this? Can be different depending on country. This first paragraph on study location can be shortened. Its adequate to talk about the medical facility, its resources, and the population being studied. It would also be good to add where geographically these facilities are located and what is the median income/ other economic variables associated with the region. Have other studies also categorized this area as underserved? Should include a citation if so.

Bolding and quotations marks are not needed ** See author formatting guidelines.

Images should be included at the end of the text and follow journal formatting guidelines. ** See author guidelines.

RESULTS

Results from the 95% confidence interval should be presented in text with odds ratios in place of the p-values. P-values can be presented in the table alongside the 95% CI and OR estimates. However, it is more appropriate to present confidence intervals when discussing the odd ratios.

DISCUSSION

How do the demographics for your study compare with the demographics living in the study area? For example, a majority of patients in this study were 51-70 years of age (no doubt due to the condition of interest). It would be interesting to know if the area served by the medical facility is located in a generally older population setting.

The discussion was well thought out and included comparison to other studies.

Reviewer #2: Pre-hospital delay and its associated factors in Acute Myocardial Infarction in a developing country

This paper complements existing literature on AMI care in LMICs. The main logistic regression analysis could be strengthened with a stronger theory-based rationale for the variables used. The authors seem to choose the variables for the model empirically, rather than from a theory of what is most important. They describe the findings after the fact — e.g., reasons for diabetes associated with more likelihood of >12 hour delay because diabetes is associated with low-symptom AMI — but they should test these ideas within the data that they have. These additional analyses don’t need their own tables, but they can be reported within the text.

The analysis should as much as possible derive from which factors are most influential for policy and practice, such as identifying which populations to be targeted.

The hours of delay as a quantity is also important due to effect of each increment of delay on outcomes: e.g., authors say each 30 minute delay before PCI associated with greater 1-year mortality. dichotomizing the hours of delay and doing logistic regression loses information, and understanding quantitative hours is also important. In addition to the logistic regression, hours of delay is a count variable, and it could be modeled with an over dispersed Poisson or negative binomial regression.

The authors very clearly justify the 12 hour dichotomized variable. This dichotomized variable isn’t a rare event, so logistic regression give odds ratios that are much larger than the relative risks. A model such as Poisson with robust standard errors or negative binomial for this binary variable would give prevalence ratios, which can be interpreted as relative risks, and will be smaller than the odds ratios.

Women have different presentations of AMI, different social roles, and table 3 indicates that the median time to presentation is 21 hours. That’s astounding. Consider separating analysis into men and women, to see whether the associations differ.

The authors describe this study as cross-sectional. However, the authors followed the patients by medical records in addition to the survey, so it would be reasonable to describe this as a retrospective cohort study. The investigators’ design is actually a very usual retrospective cohort study.

The power is well-justified for a bivariate analysis. However, the authors do a multivariate regression analysis. The paper would be stronger with some type of sample size justification. Consider Gelman and Carlin’s Type S and Type M errors, at least to estimate probability that the estimates are in wrong direction.

Sentences shouldn’t begin with a number.

Table 1 includes prior PCI and CABG, but this section seems like it should have also prior AMI as complementary information. — presumably this information was recorded and excluded as oversight?

The authors group smoking/tobacco into just one variable. Mode of ingesting tobacco may be important to report separately (e.g., bidi), and also betel/areca nut, if these were asked on survey.

Table 2 gives just a few numbers and would be much more clear as a data display because clearly there’s a long tail on this distribution on time to arrive to hospital, and it’s very right-skewed. The information could be displayed in 4 histograms, for instance.

Tables should start on their own page and not run to the next.

Table 3 has a typo: under total column, percent of women is listed as 62.4%.

Age as dichotomized variable loses so much information. 3-5 groups may be more reasonable for age so the authors have a chance to detect an age effect.

The p-value in table 3 is strictly to test the dichotomized version of the variable. However, a test of the quantitative variable would also be useful. It doesn’t need its own column: test the medians using Wilcoxon and use asterix or bold to indicate significance. I suspect 4 hour median difference by hypertension status is significant.

The authors find in table 3 that in general people with risk factors take longer to get to the hospital: people with hypertension, diabetes, family history, and smokers. Why is this? Is this association due to socioeconomic status? Age? The authors can evaluate this question with their data.

The authors put a lot of variables into their logistic regression model that have multicollinearity, such as education and income. Putting many variables into a model and seeing what is significant doesn’t actually find out what is most important. The limited sample size makes this difficult, but some variables are so important that they could be stratified on. For instance, typical vs atypical symptoms, which is the largest effect. The variables associated with longer time for the atypical symptoms may differ from the variables important for people with typical symptoms. Distance from hospital is also very important. Other variables may be more appropriate to include as numbers, such as distance from hospital and income and years of education.

The regression has diagnosis as a predictor of delay, but diagnosis happened after the delay (after going to the hospital, that is) rather than before, so it’s not logical to put it into the regression. If there is a variable that is associated with the NSTEMI vs STEMI diagnosis that is apparent prior to going to the hospital, that would be more reasonable. In discussion authors mention less severe symptoms and say they can’t test. However, the symptom related variables collected in the survey may be associated with these.

The implications of this research for policy and practice should be thought about before choosing variables. For instance, if the question is which populations should be targeted for messaging about AMI symptoms, then the variables should correspond to demographic variables and also include interaction terms: e.g., between income and gender because low-income women may have longer delays than low-income men and than women overall.

“Diabetes Mellitus is not only a major risk factor of coronary artery disease, it also results in silent MI (25) and more atypical symptoms, all of which could contribute to delays in recognizing symptoms, and hence delays in presentation.” — this is interesting, and it’s a hypothesis that could be tested with the symptom data that was collected.

6. PLOS authors have the option to publish the peer review history of their article (what does this mean?). If published, this will include your full peer review and any attached files.

Reviewer #1: No

Reviewer #2: No

---

## [Author Response · Author response to Decision Letter 0]

9 Sep 2021

Thanks a lot to both the reviewers for an excellent feedback. Apart from making my manuscript more technically sound, I also learnt a great deal from these comments. I apologize for submitting the revision this late because there has been a huge surge in COVID cases in my country the last few months, so the coauthors and I were overwhelmed with our workload in our hospitals and also had to maintain a social bubble so could not sit and work on this together. I am addressing each point of the reviewers below.

Reviewer#1 

I am very grateful to you for such a wonderful feedback. I corrected my manuscript accordingly and addressing each point below

Comment:

The authors do not discussion findings from previous studies examining pre-hospital delay for cardiovascular outcomes.

Paragraphs 2 and 3 can be combined as they are talking about the same thought. After shortening the combined paragraph more can be added which presents results from actual scientific studies as opposed to just medial guidelines.

The introduction is not sufficient as it stands. The authors need to use past studies to make a case for the issue they are studying and point out the gaps in previous research that their research addresses.

Reply: Thank you for pointing it out. Paragraph 2 and 3 of “Introduction” section have been merged and shortened. Relevant findings from other studies have also been added to make the study stand to reason. The research gap has also been highlighted.

Comment: The authors need to define mid to lower socio-economic class populations. What income level is this? Can be different depending on country. This first paragraph on study location can be shortened. Its adequate to talk about the medical facility, its resources, and the population being studied. It would also be good to add where geographically these facilities are located and what is the median income/ other economic variables associated with the region. Have other studies also categorized this area as underserved? Should include a citation if so.

Reply: We have added the income level in both Bangladeshi currency as well as US dollars. The first location has also been added and a brief insight has been given regarding the resources and population being studies. Both the hospitals are location in Dhaka, the capital city of Bangladesh. There is not enough data on the median income or other economic variables associated with this region. Moreover, both the hospitals serve patients from all parts of the country so the catchment area is very large.

Comment: Bolding and quotations marks are not needed ** See author formatting guidelines.

Images should be included at the end of the text and follow journal formatting guidelines. ** See author guidelines.

Reply: Pardon me for the silly mistake. I have now corrected the format according to guidelines

Comment: Results from the 95% confidence interval should be presented in text with odds ratios in place of the p-values. P-values can be presented in the table alongside the 95% CI and OR estimates. However, it is more appropriate to present confidence intervals when discussing the odd ratios.

Reply: Thanks for enlightening me regarding this. I have corrected the results accordingly. Odds ratios with its 95% confidence interval has been presented in both text and table with p-values alongside the OR(95% CI).

Comment: How do the demographics for your study compare with the demographics living in the study area? For example, a majority of patients in this study were 51-70 years of age (no doubt due to the condition of interest). It would be interesting to know if the area served by the medical facility is located in a generally older population setting.

Reply: This is a very important point. Unfortunately we don’t have enough data on age distribution in the population of concern.

Reviewer#2

Thank you very much to you for such details comments. Not only did all these comments make my manuscript as meticulous as I could, it also taught me a great deal as a junior doctor/researcher. Each comment you made was very valid and resonated well with our coauthors. Unfortunately, being from a low-middle income country, we are not too acquainted with some of the statistical methods. We revised the manuscript as much as we could within our capabilities. Thank you once again.

Comment: The hours of delay as a quantity is also important due to effect of each increment of delay on outcomes: e.g., authors say each 30 minute delay before PCI associated with greater 1-year mortality. dichotomizing the hours of delay and doing logistic regression loses information, and understanding quantitative hours is also important. In addition to the logistic regression, hours of delay is a count variable, and it could be modeled with an over dispersed Poisson or negative binomial regression.

The authors very clearly justify the 12 hour dichotomized variable. This dichotomized variable isn’t a rare event, so logistic regression give odds ratios that are much larger than the relative risks. A model such as Poisson with robust standard errors or negative binomial for this binary variable would give prevalence ratios, which can be interpreted as relative risks, and will be smaller than the odds ratios.

Reply: Thank you for this comment. Unfortunately, we are acquainted with binary logistic regression analysis but not acquainted with Poisson regression analysis. Also, since more than 12-hour delay is a sensitive issue for AMI patients in terms of clinical significance, we dichotomized the delay in ≤ 12 and > 12 hours to see what factors could contribute to > 12 hours delay. However, we have mentioned this issue in limitations

Comment: Women have different presentations of AMI, different social roles, and table 3 indicates that the median time to presentation is 21 hours. That’s astounding. Consider separating analysis into men and women, to see whether the associations differ.

Reply: We also thought about the different presentations of AMI between genders. We did not do a separate male and female analysis because of a relatively small sample size and also the numbers would be too small to obtain a meaningful difference between the genders. Although women with AMI had significantly longer delay in presentation to hospital in univariate analysis, they did not emerge as significant predictor in multivariate analysis. Additionally it will lengthen the result section of the study by a lot.

Comment: The authors describe this study as cross-sectional. However, the authors followed the patients by medical records in addition to the survey, so it would be reasonable to describe this as a retrospective cohort study. The investigators’ design is actually a very usual retrospective cohort study.

Reply: Thank you for this comment, another lesson learnt. We looked into the medical records for clinical presentation and comorbidities they had and did not follow the patients prospectively. We admit that the design is reasonably a retrospective cohort and changed it accordingly

Comment: The power is well-justified for a bivariate analysis. However, the authors do a multivariate regression analysis. The paper would be stronger with some type of sample size justification. Consider Gelman and Carlin’s Type S and Type M errors, at least to estimate probability that the estimates are in wrong direction.

Reply: We used the formula n=(z^2 p(1-p))/d^2 to justify the sample size. I am very sorry but we are junior researchers so we are not acquainted with Gelman and Carlin’s Type S and Type M errors.

Comment: Table 1 includes prior PCI and CABG, but this section seems like it should have also prior AMI as complementary information. — presumably this information was recorded and excluded as oversight?

Reply: This is a very valid point. We did plan to take previous Acute MI data. However, there were many patients from rural areas who aren’t well aware of their medical records and can’t provide medical records. It was hard to differentiate Acute MI or an angina episode from history alone when they did not undergo any procedure. However, they can definitely say if they underwent PCI and CABG, so we eventually decided to not use H/O Acute MI since the data would not be reliable.

Comment: The authors group smoking/tobacco into just one variable. Mode of ingesting tobacco may be important to report separately (e.g., bidi), and also betel/areca nut, if these were asked on survey.

Reply: Unfortunately, we generalized tobacco as a whole and did not separately inquire about smoking vs betel leaf consumption. I will keep this in mind for my future papers.

Comment: Table 2 gives just a few numbers and would be much more clear as a data display because clearly there’s a long tail on this distribution on time to arrive to hospital, and it’s very right-skewed. The information could be displayed in 4 histograms, for instance.

Reply: A very good point again, we have added 4 histograms accordingly also showing the skewness

Comments:

Tables should start on their own page and not run to the next.

Table 3 has a typo: under total column, percent of women is listed as 62.4%.

Reply: I apologize for these silly mistakes, these have now been corrected.

Comment: Age as dichotomized variable loses so much information. 3-5 groups may be more reasonable for age so the authors have a chance to detect an age effect.

Reply: We tried to do the analysis with 5 age groups but there was no still statistical significance so we just decided to show it as dichotomized to keep it consistent with the other variables in the table.

Comment: The p-value in table 3 is strictly to test the dichotomized version of the variable. However, a test of the quantitative variable would also be useful. It doesn’t need its own column: test the medians using Wilcoxon and use asterix or bold to indicate significance. I suspect 4 hour median difference by hypertension status is significant.

Reply: Unfortunately, to streamline the data collection process, we only collected data of the hypertension status of the patient but we do not have the quantitative data. Maybe we could focus on this and some other variables in a separate study in the future.

Comment: The authors find in table 3 that in general people with risk factors take longer to get to the hospital: people with hypertension, diabetes, family history, and smokers. Why is this? Is this association due to socioeconomic status? Age? The authors can evaluate this question with their data.

Reply: Thanks for this comment. We did not find any significant >12 hours delay with hypertension, family history and smokers. As for diabetes, we added another table to show why diabetes might have been causing this delay.

Comment: The regression has diagnosis as a predictor of delay, but diagnosis happened after the delay (after going to the hospital, that is) rather than before, so it’s not logical to put it into the regression. If there is a variable that is associated with the NSTEMI vs STEMI diagnosis that is apparent prior to going to the hospital, that would be more reasonable. In discussion authors mention less severe symptoms and say they can’t test. However, the symptom related variables collected in the survey may be associated with these

Reply: We did not include STEMI and NSTEMI in the regression.

“Diabetes Mellitus is not only a major risk factor of coronary artery disease, it also results in silent MI (25) and more atypical symptoms, all of which could contribute to delays in recognizing symptoms, and hence delays in presentation.” — this is interesting, and it’s a hypothesis that could be tested with the symptom data that was collected.

Reply: This was an excellent point. As I previously mentioned, we added an entire table showing the association of diabetes and type of presentation.

---

## [Decision Letter · Decision Letter 1]

13 Sep 2021

PONE-D-21-11112R1Pre-hospital delay and its associated factors in Acute Myocardial Infarction in a developing countryPLOS ONE

Dear Dr. Chowdhury,

Thank you for submitting your manuscript to PLOS ONE. After careful consideration, we feel that it has merit but does not fully meet PLOS ONE’s publication criteria as it currently stands. Therefore, we invite you to submit a revised version of the manuscript that addresses the points raised during the review process.

The reviewers commented favorably on your manuscript, but had some worthwhile suggestions. The authors should address the remaining issues. I am pleased to accept your manuscript, based on your revising it.

We look forward to receiving your revised manuscript.

Kind regards,

Michinari Nakamura, MD

Academic Editor

PLOS ONE

Journal Requirements:

Reviewers' comments:

Reviewer's Responses to Questions

**Comments to the Author**

1. If the authors have adequately addressed your comments raised in a previous round of review and you feel that this manuscript is now acceptable for publication, you may indicate that here to bypass the “Comments to the Author” section, enter your conflict of interest statement in the “Confidential to Editor” section, and submit your "Accept" recommendation.

Reviewer #1: All comments have been addressed

2. Is the manuscript technically sound, and do the data support the conclusions?

Reviewer #1: Partly

3. Has the statistical analysis been performed appropriately and rigorously? 

Reviewer #1: No

4. Have the authors made all data underlying the findings in their manuscript fully available?

Reviewer #1: Yes

5. Is the manuscript presented in an intelligible fashion and written in standard English?

Reviewer #1: Yes

6. Review Comments to the Author

Reviewer #1: Authors made an excellent attempt at making all reviewer suggestions. Still a few minor things remain which affect overall interpretation and reading of the manuscript.

1) P values need to be removed from OR estimate sin the abstract and the results/discussion. You can leave them in the regression table, but it is most appropriate to interpret OR along with 95% CI's.

** You will notice for example that one of your variables was significant at the 0.03 level but the interval included 1, meaning although statistically significant it could not be as influential as others which did not include 1 in the interval.

2) Did you logistic regression include two outcomes? If not, then the regression that was conducted was a multivariable logistic regression, not a multivariate. Univariate logistic regression is a fine model selection to builds a final model, but realize that including all variables in a model may change the associations observed in univariate models. SOI in this way a forward or backward variable selection process maybe more appropriate than using univariate logistic regression to identify influential variables.

** The wording for multivariate needs changed. The other comment doesn't need included in the manuscript, other than to point out in the discussion that there are potential confounders not included in bivariate models.

3) Although the authors include a paragraph which mentions previous studies, they do not present specific findings from these in the introduction. This change is needed to enhance the readers experience and understanding for the current state of myocardial infarction in that country.

7. PLOS authors have the option to publish the peer review history of their article (what does this mean?). If published, this will include your full peer review and any attached files.

Reviewer #1: No

---

## [Author Response · Author response to Decision Letter 1]

18 Oct 2021

Thanks to the reviewer for such a wonderful feedback. Once again, not only has these comments have helped this manuscript to be technically more sound, it also taught me a lot. We have attempted to revise as per the comments. I hope and pray that these revisions would suffice for the article to be eligible for publication. 

Response to Reviewers:

Comment : 1) P values need to be removed from OR estimate sin the abstract and the results/discussion. You can leave them in the regression table, but it is most appropriate to interpret OR along with 95% CI's.

** You will notice for example that one of your variables was significant at the 0.03 level but the interval included 1, meaning although statistically significant it could not be as influential as others which did not include 1 in the interval.

Response: Thanks for this comment, another lesson learnt. The p-values have been removed accordingly from the text and only the ORs with their 95% CI have been mentioned. 

Comment: 2) Did your logistic regression include two outcomes? If not, then the regression that was conducted was a multivariable logistic regression, not a multivariate. Univariate logistic regression is a fine model selection to builds a final model, but realize that including all variables in a model may change the associations observed in univariate models. SOI in this way a forward or backward variable selection process maybe more appropriate than using univariate logistic regression to identify influential variables.

** The wording for multivariate needs changed. The other comment doesn't need included in the manuscript, other than to point out in the discussion that there are potential confounders not included in bivariate models.

Response: I am sorry but as a junior researcher I am not sure whether I understood this comment correctly. Our logistic regression analysis included two outcomes so we kept the term “multivariate”. The wording for the multivariate regression analysis have been changed a little bit.

3) Although the authors include a paragraph which mentions previous studies, they do not present specific findings from these in the introduction. This change is needed to enhance the readers experience and understanding for the current state of myocardial infarction in that country.

Response: I apologize for not mentioning this. Findings have been added for reference 4 accordingly in introduction.

---

## [Editor Report · Decision Letter 2]

2 Nov 2021

Pre-hospital delay and its associated factors in Acute Myocardial Infarction in a developing country

PONE-D-21-11112R2

Dear Dr. Chowdhury,

We’re pleased to inform you that your manuscript has been judged scientifically suitable for publication and will be formally accepted for publication once it meets all outstanding technical requirements.

Kind regards,

Michinari Nakamura, MD

Academic Editor

PLOS ONE
---

## [Editor Report · Acceptance letter]

11 Nov 2021

PONE-D-21-11112R2 

Pre hospital delay and its associated factors in Acute Myocardial Infarction in a developing country 

Dear Dr. Chowdhury:

I'm pleased to inform you that your manuscript has been deemed suitable for publication in PLOS ONE. Congratulations! Your manuscript is now with our production department. 

Kind regards, 

on behalf of

Dr. Michinari Nakamura 

Academic Editor

PLOS ONE